Tryptophan-kynurenine pathway: a possible new mechanism for the prevention and treatment of reproductive system-related diseases

Ou Zhongkai 1
Xu Aixia 2
Su Hua 3
Liu Yiting 4
Li Jia lijia4199@ncu.edu.cn 4
1 The 1st Affiliated Hospital, Jiangxi Medical College, Nanchang University , Nanchang , China
2 Jiangxi Maternal and Child Health Hospital , Nanchang , China
3 HuanKui Academy, Jiangxi Medical College, Nanchang University , Nanchang , China
4 School of Basic Medical Sciences, Jiangxi Medical College, Nanchang University , Nanchang , China
Oliveira Sonia
Electronic publication date: 2025 Nov 17
Publication date: 2025
Volume: 13
Electronic Location ID: e20342
Received 2025 Apr 22; Accepted 2025 Oct 15
Copyright: ©2025 Ou et al.
Copyright year: 2025
Copyright holder: Ou et al.
License: This is an open access article distributed under the terms of the Creative Commons Attribution License, which permits unrestricted use, distribution, reproduction and adaptation in any medium and for any purpose provided that it is properly attributed. For attribution, the original author(s), title, publication source (PeerJ) and either DOI or URL of the article must be cited.
License URL: https://creativecommons.org/licenses/by/4.0/

Keywords: Molecular mechanism, Reproductive system diseases, Tryptophan-kynurenine pathway, Amino acid metabolism

Funding: The National Natural Science Foundation of China 82260293 82160284 82071708 The Major Academic and Technical Leaders Training Program of Jiangxi Province 20194BCJ22005 The Natural Science Foundation of Jiangxi province 20232BAB206022 20202ACB216003 The ‘Double-Thousand Talents Plan’ of Jiangxi Province jxsq2019201091 This work was supported by the National Natural Science Foundation of China (grant numbers 82260293, 82160284, 82071708), the Major Academic and Technical Leaders Training Program of Jiangxi Province (grant number 20194BCJ22005), the Natural Science Foundation of Jiangxi Province (grant number 20232BAB206022, 20202ACB216003), the ‘Double-Thousand Talents Plan’ of Jiangxi Province (grant number jxsq2019201091). The funders had no role in study design, data collection and analysis, decision to publish, or preparation of the manuscript.

==============================
Tryptophan (Trp) is one of the essential amino acids, and its metabolic pathway is essential for the maintenance of normal human physiological activities. Among them, various metabolites and rate-limiting enzymes of the tryptophan-kynurenine metabolic pathway play important roles in inflammatory responses, immune regulation, energy metabolism, as well as neuroprotective and toxic effects. Abnormalities in tryptophan-kynurenine pathway metabolism thus inevitably lead to numerous pathological changes, such as reproductive disorders. It is noteworthy that the prevention and treatment of reproductive diseases are not currently focused on this pathway. In view of the increasing number of studies that have found abnormalities in the levels of key enzymes and metabolites of tryptophan-kynurenine in reproductive diseases, this article will focus on summarizing the molecular mechanisms and relevance of the pathway in reproductive diseases, as well as proposing new ideas for the prevention and treatment of reproductive diseases, including the use of inhibitors of the pathway and the regulation of tryptophan metabolism in the intestinal flora.

Introduction

According to the prevalence of infertility from 1990 to 2021, up to 17.5% of adults worldwide suffer from infertility (Cox et al., 2022), posing a huge challenge to global fertility. This will undoubtedly advance the aging process of the world’s population and invariably increase the pressure on health care and social services. Therefore, it is urgent to actively prevent and treat reproductive diseases to reduce the prevalence of infertility. Currently, reproductive diseases are characterized by a variety of pathological mechanisms, but the tryptophan-kynurenine pathway appears to be a possible new mechanism for the prevention and treatment of reproductive system-related diseases.

Tryptophan (Trp) is one of the eight essential amino acids in the human body, and a small portion of Trp is metabolized to 5-hydroxytryptophan (5-HT), which is an important inhibitory neurotransmitter in the human body, but its main metabolic pathway, kynurenine pathway (KP), accounts for 95% of Trp metabolism (Badawy, 2002; Bender, 1983). KPs consist of a variety of related enzymes and metabolites with different activities. In the KP, the first step of Trp catabolism is catalyzed by tryptophan 2,3-dioxygenase (TDO) (including TDO1 and TDO2) and indoleamine 2,3-dioxygenase (IDO) (including IDO1 and IDO2), which catabolize Trp to kynurenine (Kyn), and this step is the rate-limiting step. Kyn is further catabolized mainly through two different pathways:

(1) It is catalyzed by kynurenine-3-monooxygenase (KMO) to 3-hydroxykynurenine (3-HK), which is catalyzed by kynureninase (KYNU) for further metabolism to 3-hydroxyanthranilic acid (3-HAA) and then through a multistep reaction to quinolinic acid (QUIN) and oxidized nicotinamide adenine dinucleotide (NAD) (Smith, Jamie & Guillemin, 2016).

(2) Kyn is catalyzed by kynurenine aminotransferase (KAT) to produce kynurenic acid (KYNA) (Badawy, 2017).

Key enzymes and intermediates of the tryptophan-kynurenine pathway play important roles in reproductive disorders, such as reproductive system tumors and inflammation. Clinical studies have found that high expression of key enzyme genes of the kynurenine pathway activates their endogenous ligands, which are involved in mammary tumorigenesis (Vacher et al., 2018). In addition, increased activity of key enzymes such as KMO, KYNU and KAT induced by endotoxin is involved in the inflammatory response of the body (Millischer et al., 2021). Interestingly not only the body itself, but also the tryptophan-kynurenine metabolism of the intestinal flora can be used as an entry point for reproductive disease control. Abnormalities in tryptophan-kynurenine pathway metabolism in the intestinal flora have been shown to be one of the etiologic factors for decreased ovarian function (Wang et al., 2022a). However, tryptophan-kynurenine metabolism also plays a positive role in the reproductive system. For example, testicular immune privilege and protection of the fetus in utero from immune rejection require the involvement of IDO immunomodulatory effects (Takao et al., 2007; Zhan et al., 2020). In summary, the role of the tryptophan-kynurenine pathway in the reproductive system is complex and important, and there is an urgent need to summarize the pathological roles of its key enzymes and intermediates in different reproductive diseases. A brief summary of kynurenine pathway see in Fig. 1.

Figure 1 Kynurenine pathway.

(Badawy, 2002; Bender, 1983; Smith, Jamie & Guillemin, 2016; Badawy, 2017).

Therefore, this review will systematically introduce the mechanisms of the key enzymes and important intermediates of the kynurenine metabolic pathway in the development of reproductive system-related diseases, which is crucial for elucidating the mechanisms of some reproductive system diseases and is expected to provide new ideas for the prevention and treatment of reproductive system diseases.

Audience

This review is for researchers or clinicians interested in the tryptophan-kynurenine metabolic pathway and reproductive disorders.

Survey methodology

Search engines and databases: We utilize reputable scientific databases including PubMed, Web of Science, these platforms provide access to a wide range of excellent peer-reviewed articles, ensuring coverage of keywords with relevant research across multiple disciplines. Search terms: We used specific and comprehensive search terms to retrieve relevant studies. These included: “Tryptophan-kynurenine pathway”, “kynurenine”, “reproductive system -related diseases”, ‘reproduction’, ‘treatment’, etc.. The correct combination of Boolean operators and keywords ensured that we included all relevant studies while avoiding irrelevant ones. Inclusion criteria: Articles were selected based on: Focus on the tryptophan-kynurenine metabolic pathway, in particular the key enzymes and intermediates in the tryptophan-kynurenine metabolic pathway that are involved in the mechanisms of reproductive disorders. Detailed research targeting potential control mechanisms of the kynurenine pathway. Research-based articles, case studies, reviews and meta-analyses published mainly in the last decade to ensure relevance. Exclusion criteria: Studies lacked clear conclusions or reproducible experimental methods. Literature that focused only on changes in kynurenine-related metabolites in reproductive disorders without detailing the mechanism of action on the disease or literature that mentioned a related metabolite but the mechanism of prevention and treatment did not target the metabolite. Screening process: Titles and abstracts were reviewed for relevance, followed by a detailed review of the full text methods, and conclusions. For reviews the reference lists were also reviewed to identify additional sources.

Role of enzymes and metabolites of KP in the reproductive system

IDO and the reproductive system

Indoleamine 2,3-dioxygenase (IDO) functions as an immunomodulatory enzyme. Metabolites produced by IDO1-catalyzed Trp can cause the differentiation of T cells to Treg cells, the enhancement of Treg cell activity, the apoptosis of T cells and DCs, and the blockade of NK cell activity (Favre et al., 2010), thus exerting an immune-suppressive effect that promotes the growth and spread of tumors (Heng et al., 2016). Therefore, IDO levels can be used as a prognostic criterion for diseases such as cancer. In diseases related to the reproductive system, such as ovarian plasma adenocarcinoma, IDO is a marker for poor prognosis (Takao et al., 2007).

IDO gene expression is regulated by a variety of substances, including lipopolysaccharide (LPS), TNF-α, IL-1, IL-6, IFN-γ, and the Fas agonist CH11. Among them, the most important cytokine that induces IDO is IFN-γ (Cetindere et al., 2010), and IFN-γ-dependent IDO expression is mainly achieved through the expression of the STAT1 pathway. STAT1 is phosphorylated by receptor-associated kinase to form a dimer, which is translocated to the nucleus to act as a transcriptional activator and binds to the IDO gene to directly induce IDO transcription or through the IRF-1 protein to indirectly induce IDO transcription (Robinson, Hale & Carlin, 2005). In a non-IFN-γ-dependent pathway, interleukin 6 (IL-6) stimulates STAT3 phosphorylation and NF-κb-inducing kinase (NIK) expression in myeloid-derived suppressor cells (MDSCs), which mediates the upregulation of IDO expression through STAT3-dependent NF-κB (Yu et al., 2014). Increased STAT3 phosphorylation and NIK expression are associated with the upregulation of IDO expression in breast cancer in situ (Yu et al., 2014). Upon binding of CH11 or TNF-α to death receptors on the cytosolic membrane, the ASK1-JNK and IKKα-NF-κB signaling pathways are activated, which enhances the activity of the transcription factors AP-1 and NF-κB, respectively (Cetindere et al., 2010), and induces the expression of IDO genes, and the elevated level of IDO leads to the accumulation of intracellular reactive oxygen species (ROS), thereby triggering apoptosis in cells, including immune cells (Cetindere et al., 2010).

IDO exerts multiple effects on the reproductive system as well as pregnancy. It has been shown that IDO is associated with cancer cells and the cancer microenvironment. IDO1 may act as an immunomodulatory factor that contributes to the endometrial cancer microenvironment (Zhan et al., 2020), and endometrial cancer cells have greater IDO1 activity than noncancerous endometrial cells (Zhou et al., 2021), emphasizing the importance of determining IDO1 activity in cancerous tissues. Overexpression of IDO in endometrial cancer cells may contribute to tumor development in vivo by inhibiting NK cell activity to exert an immunosuppressive effect (Zhan et al., 2020), suggesting that IDO is a novel and reliable prognostic indicator for endometrial cancer. Similarly, IDO can be used as a good prognostic marker for poor survival of ovarian plasma adenocarcinoma, and the amount of IDO synthesized is positively correlated with decreased survival in patients with ovarian plasma carcinoma (Takao et al., 2007). In ovarian cancer, IDO plays a role in inhibiting the proliferation and killing of NK cells as well as effector T cells, and high expression of IDO in tumor cells is associated with a decrease in the number of lymphocytes, tumor infiltration and peritoneal spread but has no effect on the growth of cancer cells (Wang et al., 2012). Moreover, IDO and immunosuppressive cytokines in ascites enhance each other, thereby promoting the metastatic spread of ovarian tumors in the peritoneum (Wang et al., 2012). In cervical cancer, IDO1 is involved in the self-renewal and expression of OCT4 and SOX2 in cervical cancer cells (Low et al., 2020), and the results of a study of a mouse model of cervical cancer suggested that inhibition of IDO1 expression reduces the size of cervical tumors and that the expression level of IDO1 is correlated with cervical cancer survival and prognosis (Low et al., 2020). IDO is also involved in endometriosis. Increased IDO expression in the ectopic endometrium induces the secretion of IL-33, which induces fine macrophage conversion to the M2 type (Mei et al., 2014). In a mouse model of endometriosis, M2-type macrophages induced angiogenesis at the lesion site (Bacci et al., 2009), which may play a major role in endometrial proliferation and is closely related to the development of endometriosis (Mei et al., 2014).

IDO can be characterized as a “double-edged sword” that, in addition to producing a variety of pathological effects, may also exert beneficial biological effects on the reproductive system. Testicular immune privilege also plays an important role in the effect of IDO in addition to isolation by the blood−testis barrier. IDO performs two main functions: one is to consume Trp in closed environments, such as the epididymal lumen, to prevent invasion of bacterial or viral infections (Romani et al., 2008), and the other is to inhibit the autoimmune response of T cells, induce an increase in the number of Treg cells, and maintain testicular immune privilege and tissue immune tolerance (Gualdoni et al., 2019). The protective role of testicular support cells is also related to the efficient metabolism of Trp by IDO (Fallarino et al., 2009). In addition, IDO activity modulates the involvement of the epididymal proteasome in the clearance of defective spermatozoa (Jrad-Lamine et al., 2011). Trophoblast invasion of endometrial tissue is a prerequisite for normal pregnancy, a process that requires the involvement of IDO (Zong et al., 2016). The extravillous trophoblast (EVT) is the embryonic cell that is in closest contact with the maternal immune system, and it has been shown that EVT is strongly positive for IDO and protects the fetus from rejection by suppressing maternal local T cells (Hönig et al., 2004; Sedlmayr, 2007). Overall, the function of IDO in the reproductive system is complicated. The following figure provides a brief summary (Fig. 2). The immunomodulatory function of IDO can also play a positive role in the reproductive system. Supplementation with exogenous transforming growth factor-β1 (TGF-β1) induces IDO expression thereby establishing maternal immune tolerance, which is important for preventing recurrent miscarriage (Wen et al., 2025). In addition, induction of IDO overexpression by vector injection into mice with premature ovarian failure revealed an increase in the proportion of Treg cells, an increase in the level of FoxP3 mRNA expression in Treg cells, and an improvement in ovarian function (Hu et al., 2025).

Figure 2 The pathological effects of IDO in the reproductive system.

(Favre et al., 2010; Takao et al., 2007; Yu et al., 2014; Zhan et al., 2020; Zhan et al., 2020; Takao et al., 2007; Wang et al., 2012; Low et al., 2020; Mei et al., 2014; Gualdoni et al., 2019).

TDO and the reproductive system

Tryptophan 2,3-dioxygenase (TDO) is mainly expressed in the liver, but it can play an important role in certain reproductive system-related diseases. TDO promotes tumor cell migration and invasion through the TDO2-Kyn-AhR pathway (Li et al., 2020). The level of TDO2 expression is positively correlated with the expression level of genes critical for uterine smooth muscle tumor growth, and the concentration of TDO2 is significantly elevated in uterine smooth muscle tumors with MED12 mutations, which further promotes tumor growth (Hutchinson et al., 2022; Zuberi et al., 2023). In a study of triple-negative breast cancer (TNBC), NF-κB activity was significantly increased in TNBC cell suspensions, and TNBC cells activated the TDO2-AhR pathway via NF-κB to promote TNBC cell proliferation and migration (D’Amato et al., 2015). In addition, elevated expression of TDO2 in the prostate cancer cell lines LNCaP and VCaP promoted Trp metabolism, and tumor cells enhanced the transport of Trp transport proteins through the activation of the NF-κB-dependent AhR/C-Myc/ABC-SLC transport protein signaling pathway, which further promoted Trp metabolism catalyzed by TDO2 in prostate cancer cells and promoted Trp metabolism in an AhR/c-Myc-dependent manner to promote chemotherapy resistance in prostate cancer (Li et al., 2021). In addition, TDO2 is upregulated in ovarian cancer tissues and promotes the proliferation and migration of ovarian cancer cells. A related study revealed that the concentration of TDO2 mRNA in immortalized ovarian surface epithelial cells (immortalized T29 cells) was >100-fold greater than that in normal T29 cells (Zhao et al., 2021). Moreover, TDO2 is correlated with the stage, recurrence and survival of patients with ovarian cancer (Tothill et al., 2008). TDO expression in stromal leukocytes surrounding invasive cervical lesions has been detected in studies of HPV-associated cervical lesions, but a definitive relationship between TDO and HPV-associated cervical lesions still cannot be determined, and further studies are needed (Venancio et al., 2019).

In addition to various types of reproductive tumors, TDO has been associated with preeclampsia (PE). In a comparison of TDO mRNA expression in the placenta of PE patients and healthy controls, it was found that TDO mRNA expression was elevated in the PE group and was associated with tryptophan depletion (Keaton et al., 2019). However, the pathologic mechanisms need to be investigated further.

Kyn and the reproductive system

Kynurenine (Kyn) is the first product of IDO/TDO-catalyzed tryptophan production, so the Kyn:Trp ratio can be used to reflect IDO/TDO activity. Kyn is a key mediator in the Trp-AhR pathway mentioned in the above article, where tryptophan is catalyzed by IDO/TDO to produce Kyn, which undergoes a related enzymatic reaction that produces derivatives and thus acts on the aryl hydrocarbon receptor (AhR) (Seok et al., 2018). However, it has also been suggested that Kyn is not a direct effector of AhR and that Trp deprivation is the real initiator of AhR (Solvay et al., 2023). But regardless of the mechanism, the activation of AhR induces CD39 expression in tumor-associated cells, promotes CD8+ T-cell dysfunction (Takenaka et al., 2019), and promotes Treg cell proliferation to induce immunosuppression (Mezrich et al., 2010), thereby enabling the tumor microenvironment. Aromatase (including CYP1A1 and CYP1B1) is the key enzyme to convert the androstenedione to estrone, which plays a significant role in reproductive diseases such as endometriosis and nonobstructive azoospermia (Kalantari et al., 2025; Shiraishi, Oka & Matsuyama, 2021). Its pathogenesis is related to the Kyn pathway. Mammary epithelial cells can take up Kyn produced by adipocytes, which induces the production of CYP1B1 and activates AhR, contributing to the malignant transformation of mammary epithelial cells leading to triple-negative breast cancer (TNBC), but this process is not important for the proliferation of TNBC (Diedrich et al., 2023). Similarly, TDO2-induced generation of Kyn can be associated with prostate cancer by acting on AhR (Li et al., 2021). Exposure to phthalates (MEHHP) is a high risk factor for uterine smooth muscle tumor growth, and MEHHP can upregulate the expression of two prototypical targets of AhR, CYP1A1 and CYP1B1, and activate the Trp-kyn-AhR pathway to reduce apoptosis to promote the survival of uterine smooth muscle tumor cells (Iizuka et al., 2022). Endometriosis (ET) tissues were enriched in ligands related to IDO1, Kyn, and AhR, as well as abundant expression of AhR on mast cells (MCs), which was hypothesized to act on MCs to release soluble factors to promote endometrial growth through the Trp-Kyn-AhR pathway, further leading to the development of ET (Mariuzzi et al., 2016). The development of polycystic ovary syndrome is also associated with the Kyn-AhR pathway. Kyn activates the AhR-PCSK9 pathway leading to abnormal lipid metabolism, elevated plasma low-density lipoprotein (LDL-C) levels, and induced ovarian dysfunction, resulting in PCOS-like pathology (Wang et al., 2025). Besides, tryptophan depletion upregulates the immune checkpoint PD-L1, increased tumor-associated macrophage infiltration, and promoted the formation of tumor microenvironment (TME) (Crump et al., 2024), which is an important mechanism for tumor progression. Inhibition of the Kyn metabolic pathway to reduce Trp depletion could also be an important potential target for cancer therapy (Lu et al., 2025). However, given that the efficacy of Kyn metabolic pathway inhibitors for tumor treatment is not outstanding, for example, the clinical trial of epacadostat combined with pembrolizumab for uroepithelial cancer did not achieve better benefits in the experimental group (Necchi et al., 2024), the specific mechanism of targeting Trp depletion for the treatment of reproductive system-related tumors needs to be further explored. In addition to the mechanisms described above, targeting the metabolism of Trp toward serotonin (5-HT) is also an important mechanism that cannot be ignored. The serum Kyn/Trp ratio is significantly higher in patients with primary premature ejaculation (PPE) compared to the healthy population, whereas the Kyn/Trp ratio was significantly decreased after treatment of PPE with selective 5-HT reuptake inhibitors (dapoxetine) (Wu et al., 2024). The possible mechanism is to target the increase in Trp metabolism from Kyn to 5-HT and inhibit 5-HT reuptake, thereby increasing central 5-HT concentration and prolonging ejaculation time.

The role of Kyn in other reproductive system-related diseases is not yet well understood, but with the available research data, the Trp-Kyn-AhR pathway and Trp depletion are currently a promising line of research for the treatment of related diseases.

KMO and the reproductive system

Kynurenine 3-monooxygenase (KMO) is another key enzyme in KP and is located on the outer mitochondrial membrane (Maddison et al., 2020). KMO catalyzes kynurenine to produce the kynurenine derivative 3-hydroxykynurenine (3-HK), which has been shown to be effective in inhibiting T-cell proliferation and is thus involved in immunosuppression (Lassiter et al., 2021) and may be associated with the development of certain tumors. In a related study, KMO was found to be highly expressed in both the cytoplasm and cell membrane of clinical breast cancer cells. Researchers used cBioPortal to analyze the relationship between alterations in KMO gene amplification, mutation, and fusion and the relationship with cancer and found that the overall survival rate in clinical breast cancer patients with KMO gene alterations was lower than that in patients without KMO gene alterations and that high KMO expression was significantly correlated with shorter overall survival in breast cancer patients (Lai et al., 2021). In addition, the migration and invasion of MDA-MB-231 cells were significantly reduced after blocking KMO, suggesting that KMO may promote the growth and metastasis of breast cancer cells (Lai et al., 2021). In a study of triple-negative breast cancer, KMO was found to regulate pluripotent genes through β-linker proteins and to exert oncogenic effects in TNBC (Huang et al., 2020).

KYNU and the reproductive system

Kynureninase (KYNU) is a key enzyme that catalyzes the formation of 3-hydroxy-o-cyanobenzoic acid from 3HK. A study on breast cancer (BC) indicated a strong positive relationship between KYNU and T-cell immune response-related gene ontologies, such as regulation of the T-cell receptor signaling pathway, suggesting that KYNU may play an inhibitory role in T-cell immunity in BC (Li et al., 2023), which may serve as an indicator of poor prognosis in BC patients. A study of relevant data revealed that a high concentration of KYNU was associated with poorer overall survival, recurrence-free survival, and distant metastasis-free survival (Li et al., 2023). The interaction between adenosine deaminase acting on RNA 1 (ADAR1) and KYNU might contribute to TNBC in MDA-MB-231 cells (Binothman et al., 2023). Another study showed that KYNU was associated with the activation of the PI3K/AKT and RAS pathways; the PI3K/AKT and RAS pathways are involved in the regulation of physiological functions such as cell migration, invasion and survival (Al-Mansoob et al., 2021), suggesting a metastatic role for KYNU in BC. Surprisingly, however, a recent study on BC has shown that tumor cell growth appears to be associated with the absence of KYNU. The results of the experiment showed that the degree of KYNU expression was negatively correlated with the degree of malignancy of BC tumors as well as the breast cancer tumor stage (T stage), i.e., KYNU was expressed at a very high level in T0-stage BC, higher in T1-stage BC, and lower in T2-stage BC; KYNU was highly expressed in hyperdifferentiated I-stage (low degree of malignancy) BCs, while in hypodifferentiated III-stage (high degree of malignancy) BCs, KYNU showed a very low expression level (Liu et al., 2019). In addition, other studies have reached conclusions that contradict the aforementioned role of KYNU: KYNU acts as a tumor suppressor in cancer, inhibiting the proliferation and development of tumor cells (Lauvrak et al., 2013; Wang et al., 2019). The above contradictory conclusions indicate that the mechanism of action of KYNU is not yet well understood, and further studies are needed. KYNU has also been associated with chemoresistance in cervical adenocarcinoma (ADC). Cervical adenocarcinoma cisplatin-resistant cells (HeLa/DDP) showed greater KYNU mRNA expression than HeLa cells, suggesting that KYNU was overexpressed in HeLa/DDP cells (Zhang et al., 2023). Knockdown of KYNU mRNA significantly decreased invasion and proliferation and significantly increased apoptosis in HeLa/DDP cells in response to cisplatin. The results showed that KYNU was involved in chemoresistance in ADC (Zhang et al., 2023). A study of uterine leiomyomas revealed that NAD levels were reduced in uterine leiomyoma tissues (Chuang et al., 2022). Since the reaction catalyzed by KYNU is an upstream reaction for the generation of NAD, it was hypothesized based on the experimental results that a decrease in KYNU expression might lead to a decrease in NAD production in leiomyomas. A decrease in NAD may lead to an increase in oxidative stress in leiomyomas, further damaging the cells (AlAshqar et al., 2023).

QUIN, QPRT and the reproductive system

Quinolinic acid (QUIN) is a product of 3-hydroxyphthalamic acid catalyzed by 3-hydroxyphthalamic acid dioxygenase (3-HAO), and QUIN is further catalyzed by quinolinic acid phosphoribosyltransferase (QPRT) to generate one of the end-products of KP, NAD; QRPT is the rate-limiting enzyme for the de novo synthesis pathway of NAD. Studies have shown that increasing NAD levels can effectively respond to oxidative stress to alleviate ovarian senescence and improve oocyte quality and fertility (Liang et al., 2023). However, in a study of invasive breast cancer, it was noted that increased QPRT expression in invasive breast cancer promotes the migration and invasion of cancer cells through purinergic signaling (Liu et al., 2020), the mechanism of which may contribute to the development and progression of breast cancer through the PI3K/AKT signaling pathway and is correlated with poorer overall survival (Yan et al., 2023; Zhou et al., 2022). In addition, the upregulation of QPRT expression has also been shown to be involved in resistance to cisplatin chemotherapy in ovarian cancer through the PI3K/AKT signaling pathway (Niu et al., 2020). Therefore, care should be taken to avoid over increasing QPRT expression by upregulating the NAD de novo synthesis pathway when administering drugs to increase the body’s NAD content. The HAAO gene encodes 3-HAO-catalyzed QUIN, and HAAO has been associated with certain reproductive system-related diseases. The HAAO allele HAAO rs386183, which can be regarded as a risk allele, has been found to have a strong correlation with the risk of anterior/middle hypospadias of the urethra in Han children in southern China in a recent study (Liu et al., 2022). However, the specific mechanism underlying the correlation between HAAO rs386183 and anterior/middle hypospadias remains unclear. Silencing HAAO expression is associated with endometrial cancer. Researchers found that the promoter of HAAO was highly methylated in 63% of endometrial cancer samples and that the methylated promoter resulted in the absence of HAAO expression (Huang et al., 2010). Furthermore, the methylation level of HAAO was significantly correlated with disease-free survival in patients with endometrial cancer (P = 0.034) (Huang et al., 2010). QUIN has been shown to have some toxic effects on the nervous system, e.g., in relation to the neurodegenerative disease Parkinson’s disease. Its toxic effects on the nervous system can indirectly affect reproductive health. Continuous injection of QUIN into rats at a concentration of 500 nmol/2 μ l has been found to have toxic effects on the arcuate nucleus of the brain, thereby affecting the normal function of the hypothalamic-pituitary-gonadal axis and the estrous cycle of female rats (Abbasi et al., 2025). However, the studies on the direct effects of QUIN on the reproductive system are not yet mature and need to be further explored. However, KYNU plays an active role in embryo attachment. KYNU shows high expression in the endometrial receptive phase, suggesting that its core role is to participate in the assembly of membrane structure and intercellular junctions, thus creating suitable conditions for embryo adhesion (Evans et al., 2020). Therefore, KYNU can be a potential target for improving fertility and treating infertility.

KAT, KYNA and the reproductive system

Kynurenine aminotransferase (KAT) catalyzes Kyn to proceed in the other direction, generating another end product of KP: kynurenine acid (KYNA). KAT isozymes exhibit completely opposite expression levels in different tumor tissues. For example, KAT3, also known as cysteine-conjugated β-cleavage enzyme (CCBL2), was found to be abundantly expressed in the human normal breast epithelial cell line MCF-10A, whereas its expression in the BC cell line was lower than that in the normal breast group, and the same finding was also found in ovarian cancer. High CCBL2 expression is associated with increased survival and recurrence-free survival (Meng et al., 2022). The possible reason is that when CCBL2 expression is low, there is aberrant activation of the proto-oncogene MYC in BC cells, and upregulation of the MYC pathway drives cancer; altered glutamine metabolism in MYC-driven BC results in glutamine addiction, which leads to decreased survival (Chen & Cui, 2015; Le et al., 2012). In contrast, KAT2 was found to be overexpressed and KYNA production was increased in uterine leiomyoma tissues (Chuang et al., 2022), which was speculated to be related to the immunosuppressive effect of KYNA (Moroni et al., 2012). KYNA-mediated immunosuppression, as well as anti-inflammatory effects, can be achieved by targeting G protein-coupled receptor 35 (GPR35) and the AhR-associated signaling pathway (Wirthgen et al., 2017). KYNA is a major metabolite of the Trp-AhR pathway that activates AhR. KYNA is a major metabolite in the Trp-AhR pathway that activates AhR. IL-6 mRNA expression is highly dependent on AhR expression, so KYNA can mediate IL-6 production and play a role in tumors through activating AhR (Walczak et al., 2020). In addition, KYNA-mediated IL-6 induces IDO expression to enhance KP activity, resulting in positive feedback regulation (See Fig. 3).

Figure 3 Positive feedback regulation of KYNA.

(Walczak et al., 2020).

KYNA has been shown to have a protective effect on the nervous system (Venkatesan et al., 2020), and some studies have shown that KYNA has similar effects on the reproductive system. For example, KYNA can inhibit inflammatory responses by inhibiting the NF-κB pathway and increasing the GRP35 signaling pathway to prevent bacterial lipopolysaccharide-induced endometritis (Wang et al., 2022c). Mechanistically, KYNA inhibits the production of TNF-α and IL-1 in LPS-induced mouse endometrial epithelial cells and upregulates tight junction proteins to maintain the permeability of the epithelial barrier, which significantly inhibits neutrophil infiltration and pathological uterine damage. Higher levels of KYNA are also associated with a lower risk of breast cancer (Zeleznik et al., 2021), but this is contradicted by the fact that the Trp-AhR pathway promotes tumor cell growth (mentioned in Section ‘TDO and the Reproductive System’). At present, the relationship between KYNA and cancer is still unclear in the academic community, and further research is needed to explore this topic. Some studies have shown that KYNA also plays a role in the occurrence of reproductive system-related diseases. For example, a study revealed that the content of KYNA was greater in the endometrium of female dogs with uterine empyema, suggesting that KYNA may be a marker of uterine empyema (Dabrowski et al., 2013). However, it is worth noting that the difference in the content of KYNA in the endometrium of female dogs with uterine empyema and in the endometrium of normal female dogs in this study was not statistically significant, which contradicts the conclusion of the anti-inflammatory effect of KYNA.

Multiple substances in KP work together

The abnormal expression of KP is not limited to one substance in KP-associated reproductive disorders but may also be the result of a combination of substances. In one study, KP was found to be abnormally activated in women with polycystic ovary syndrome (PCOS), with enhanced enzyme activities such as IDO, TDO, and KAT and elevated plasma Kyn, KYNA, and QUIN concentrations. Wang et al. (2022b) suggested that multiple key KP enzymes or intermediates may contribute to KP-related reproductive disorders. A considerable amount of literature suggests that PCOS is associated with chronic inflammation and obesity. Researchers inducing endotoxemia and cellular inflammation with LPS found increased activity of KMO, KYNU, and KAT (Millischer et al., 2021), while another study showed that KP is activated in obesity, with elevated expression of IDO-1, KYNU, and KAT II (CCBL2) in the adipose tissue of obese women (Favennec et al., 2015). The results of these two studies indicate that inflammation and obesity lead to the upregulation of the activities of multiple key enzymes in the KP, which is consistent with the results of the upregulation of the activities of multiple key enzymes in the KP in PCOS and further suggests the relevance of chronic inflammation and obesity to PCOS and the possibility of the pathogenesis of PCOS as a result of the joint action of multiple substances in the KP. However, the related mechanisms still need to be further explored. In addition, in vivo experiments in animals have demonstrated that subcutaneous implantation of hydrogels mixed with IDO1 inhibitors NLG919 and KYNU can remodel the tumor immune microenvironment through the combination of dual pathways of decreased Kyn production and increased Kyn consumption, and successfully inhibit the recurrence and metastasis of postoperative breast tumors (Liu et al., 2025). Overall, the combined modulation of the Trp-kyn metabolic pathway through multiple pathways is a promising therapy for reproductive diseases, but more clinical evidence is needed for its implementation. (Key metabolites and enzymes and their effects across various reproductive diseases are listed in Table 1).

Table 1 The roles of KP enzymes/metabolites in reproductive diseases.

A tabel which summarize KP enzymes/metabolites, their roles in specific diseases, and associated therapeutics.

KP enzymes/ metabolites	Roles in diseases	Potential therapeutics	
IDO	Breast cancer (Yu et al., 2014), endometrial cancer (Zhou et al., 2021), ovarian plasma adenocarcinoma (Takao et al., 2007), cervical cancer (Low et al., 2020), endometriosis (Mei et al., 2014)	Inhibitors (Ma et al., 2020), inhibitors combined with immunotherapy (Sieviläinen et al., 2022) inhibitors (Oweira et al., 2022), inhibitors combined with immunotherapy (Zuberi et al., 2023)	
TDO	Uterine smooth muscle tumor (Iizuka et al., 2022), breast cancer (TNBC) (D’Amato et al., 2015), prostate cancer (Li et al., 2021), ovarian cancer (Zhao et al., 2021), pre-eclampsia (Keaton et al., 2019)		
Kyn	Uterine smooth muscle tumor (Iizuka et al., 2022), breast cancer (TNBC) (Diedrich et al., 2023), PCOS (Wang et al., 2025) endometriosis (Mariuzzi et al., 2016), reproductive aging (Dai et al., 2024); nonobstructive azoospermia (Shiraishi, Oka & Matsuyama, 2021), decreased ovarian function (Wang et al., 2022a)	Inhibit Kyn-AhR pathway (Liao et al., 2025), regulate gut flora’s kyn metabolism (Wang et al., 2022a)	
KMO	Breast cancer (TNBC) (Huang et al., 2020)	——	
KYNU	Breast cancer (Li et al., 2023), cervical cancer (Zhang et al., 2023)r	——	
QPRT	Breast cancer (Liu et al., 2020), ovarian cancer (Niu et al., 2020)	——	
KAT	Breast cancer (Meng et al., 2022), uterine smooth muscle tumor (Chuang et al., 2022)	——	

Possible Mechanisms of KP in the Treatment of Reproductive Diseases

A large number of studies have demonstrated the correlation between the kynurenine pathway and the development of tumors or other diseases; thus, targeting the kynurenine pathway for the treatment of related diseases has attracted much attention. Considering the relationship between reproductive system-related diseases and the kynurenine pathway as described in the article, the following section will focus on the regulation of KP and discuss the potential mechanisms for the prevention and treatment of reproductive system-related diseases.

Regulation of key enzyme activities of KP

Regulation of IDO activity

Given that the product of Trp catalyzed by IDO has immunosuppressive effects, promotes the formation of the tumor microenvironment as well as the migration and proliferation of tumor cells, and has been shown to be abundantly expressed in tumors related to the reproductive system, the use of IDO inhibitors in this disease has some palliative and therapeutic effects. The most reported IDO inhibitor in relevant studies is 1-methyltryptophan (1-MT), also known as Indoximod. Indoximod significantly inhibits tumor-derived IDO, thereby enhancing the activity of tumor-infiltrating effector T cells (Brincks et al., 2020), which was found to significantly inhibit ascites and peritoneal spreading of ovarian tumors, reducing the secretion of transforming growth factor-β (TGF-β) (Nonaka et al., 2011), and activating immune cell function while decreasing the expression of IDO to inhibit the metastasis of carboplatin-resistant ovarian cancer cells (Ma et al., 2020). IDO has also been found to be overexpressed in cancer tissues such as those of paclitaxel-resistant ovarian cancer and is associated with drug resistance (Fujiwara et al., 2024). However, indoximod enhances the activity of standard chemotherapeutic agents. For example, pembrolizumab combined with indoximod was well tolerated and active in a pretreated population of patients with advanced melanoma (Zakharia et al., 2021). In addition, in the MMTV-Neu mouse model of breast cancer, indoximod in combination with paclitaxel was found to reduce the MMTV-Neu breast tumor volume by 30% after two weeks of treatment in mice (Muller et al., 2005), and indoximod also had an enhancing effect on cisplatin, doxorubicin, and cyclophosphamide (Muller et al., 2005). Currently, the combination of IDO inhibitors with immunotherapy is a promising therapeutic direction, and numerous clinical trials, such as NCT03343613, have made some progress in the study of IDO inhibitors combined with immunization against PD-L1 checkpoints in the treatment of solid tumors, such as ovarian cancer, breast cancer, etc. Epacadostat, a selective IDO1 inhibitor, has been studied in a variety of malignancies and in combination with other immunosuppressive checkpoint agents, including ipilimumab (anti-CTLA-4 antibody) (Gibney et al., 2019) and nabulizumab (anti-PD-1 antibody) (Sieviläinen et al., 2022). Among them, epacadostat in combination with nabulizumab for the treatment of advanced ovarian cancer showed an antitumor effect, with a disease control rate of 28% (Gibney et al., 2019). In addition, a phase II clinical trial found that epacadostat in combination with pembrolizumab resulted in an overall remission rate of 21% in patients with recurrent clear cell carcinoma of the ovary (Gien et al., 2024), demonstrating the therapeutic promise of an IDO inhibitor in combination with immunotherapy. Despite their potential therapeutic capabilities, IDO inhibitors have certain safety concerns. These include toxicity of the inhibitor itself and adverse effects of inhibiting IDO outside of the target tissue. IDO acts as an immunomodulator, and inhibition of its activity produces a range of immune adverse events, such as immune-mediated liver injury, adrenocortical insufficiency, etc. (Yoshimura et al., 2023). Adverse effects due to IDO itself are similar to those seen with classical immunotherapy, and include itching of the skin, skin rashes, and gastrointestinal reactions (Le Naour et al., 2020). Therefore, IDO inhibitors should be critically evaluated for adverse immune reactions outside the target organ when applied. Furthermore, despite the use of IDO inhibitors to suppress tumors or other diseases, degrading IDO at the site of the lesion directly is another alternative approach. For example, ubiquitination-mediated degradation of IDO1 was achieved using protein hydrolysis-targeted chimeric technology (Hu et al., 2020). However, inhibition or degradation of IDO should also be performed with attention to its positive effects in certain tissues and organs, e.g., it should be avoided to disrupt the immune privilege of the testis and affect cells such as spermatozoa. Instead, it is necessary to enhance IDO activity and modulate the immune function of the body in the control of certain reproductive diseases. Examples include delaying premature ovarian failure and the treatment of recurrent miscarriages (see in Section ‘IDO and the Reproductive System’). In addition, by analyzing plasma metabolites in aged and young rats, it was found that kynurenine levels decreased in aged rats compared to the young group. Plasma kynurenine levels could be increased in the aged group by acupuncture therapy, which repaired the ability of neurotransmitters and gene transcription, thus restoring the function of the hypothalamic-pituitary-ovarian axis to alleviate neuroendocrine ovarian senescence (Dai et al., 2024). Although this study did not directly analyze the mechanisms by which acupuncture therapy improves kynurenine metabolism and the correlation between kynurenine and neuroendocrine aging, it provides us with new ideas for improving kynurenine metabolism (e.g., modulating IDO activity) to intervene in reproductive aging. The specific role of the kynurenine pathway in neuroendocrine ovarian aging needs to be explored in depth in the future.

Inhibitors of TDO

Similar to that of IDO, TDO expression is significantly upregulated in a variety of tumors, so the use of TDO inhibitors may also be useful for treating reproductive system-related tumors. In studies of uterine smooth muscle tumor cells (LM), the selective TDO inhibitor 680C91 was found to knock down TDO2 (Chuang et al., 2021; Chuang et al., 2024) and significantly downregulate CYP1B1 gene expression (Chuang et al., 2024); this effect can reduce AhR levels and inhibit the role of TDO2 in promoting LM growth. Certain hormones can also inhibit the biological activity of TDO. The use of progesterone agonists was found to reduce TDO2 expression in smooth muscle tumor cells with a mutation in the MED12 gene (Hutchinson et al., 2022), suggesting that progesterone may act as an inhibitor of TDO2 and be involved in the anti-smooth muscle tumor effect in the uterus. LM10, a TDO inhibitor, has been shown to be effective in inhibiting the growth of TDO-expressing tumors (Oweira et al., 2022). TDO2 can be used as a new target for the treatment of ovarian cancer (as mentioned above), and the use of LM10 in the treatment of ovarian cancer reduces the level of TDO2 expression and inhibits the proliferation, migration, and invasion of ovarian cancer cells (Zhao et al., 2021) to produce an antitumor effect. miRNA-200c was found to be expressed at low levels in TNBC cell lines (Niedźwiecki et al., 2018). Researchers restored the expression of miRNA-200c in TNBC and found that Kyn production was reduced in TNBC. Further experiments demonstrated that miRNA-200c can directly target TDO2, thereby reducing Kyn production in TNBC cell lines (Rogers et al., 2019). In addition, miRNA-200c inhibited many genes encoding immunosuppressive factors, including the death receptors PD1/2, HMOX1, and GDF15 (Rogers et al., 2019). The anticancer effect of the TDO inhibitor miRNA-200c was demonstrated. Heat stress damages granulosa cells and thus affects ovarian function. It was found that elevated TDO2 expression is importantly associated with heat stress, mediating tryptophan degradation in heat stress. Vitamin C supplementation can effectively alleviate abnormal tryptophan metabolism and thus improve ovarian function (Sammad et al., 2024). Therefore, it is hypothesized that TDO inhibitors can also be used for ovarian dysfunction mediated by similar mechanisms, but further clinical trials are needed to validate the results. Currently, studies on TDO inhibitors are in the preclinical stage, and no approved products have yet been marketed. Therefore, there is still a great challenge to apply TDO inhibitors in the clinic.

Inhibitors of KMO

Currently, research on KMO inhibitors has been limited to their neurological effects, with few evaluations of their therapeutic use in cancer or other diseases and even less research in the reproductive system. Given that KMO has been shown to be associated with breast cancer cell migration and proliferation and patient survival in breast cancer, it is hypothesized that KMO inhibitors may play a positive role in reproductive system-related cancers or other diseases. Several KMO inhibitors are listed below. Sulfonamides are among the most potent KMO inhibitors known. For example, Ro-61-8048 blocks KMO in autologous multiple myeloma cells, produces potent CD8+ CTL activity and significantly increases the killing function of NK cells and attenuates cancer cell migration and invasion (Ray et al., 2020). Ro-61-8048 increases KYNA production while blocking KMO (Moroni et al., 2005). Oral administration of Ro-61-8048 at 42 mg/kg increased KYNA concentrations in the hippocampus sevenfold in rats (Drysdale et al., 2000). However, some scholars have found in studies on breast cancer that inhibition of KP with Ro-61-8048 suppressed the polarization of M1 macrophages (Xue et al., 2023), which are antitumor-type macrophages (Gao, Liang & Wang, 2022). The experimental results are contrary to these findings and still need to be further explored. The variability in the results is hypothesized to be related to the site of KMO inhibitor administration as well as the concentration. The KMO inhibitor UPF648 has a similar effect to that of Ro-61-8048, decreasing QUIN concentrations and increasing KYNA concentrations, which is neuroprotective. However, at the same time, it significantly increased the production of hydrogen peroxide by nearly 20-fold (Hughes et al., 2022), which may cause oxidative stress or certain other lesions in the cells. The only clinical information available on KMO inhibitors comes from a clinical trial of the KMO inhibitor GSK3335065 in a healthy population. The study found that its most common adverse event was headache, while serious adverse events included ventricular arrhythmias, but were ultimately judged to be unrelated to the drug itself (Fernando et al., 2022). Thus, further research on KMO inhibitors in the treatment of reproductive diseases remains to be done.

Metabolite interventions of KP

Inhibition of the Kyn-AhR pathway

The aryl hydrocarbon receptor (AhR) is a transcription factor that has been linked to a variety of reproductive disorders (see ‘Role of enzymes and metabolites of KP in the reproductive system’ for further details). Kyn has been identified as an endogenous ligand for the AhR, thereby activating the receptor and inducing a series of biological effects (Shadboorestan et al., 2023). The underlying pathogenic mechanism may be associated with the induction of an imbalance between autophagy and apoptosis. Kyn has been shown to upregulate the activity of the Kyn/AhR/mTOR axis, promote the phosphorylation and cytoplasmic retention of the transcription factor EB (TFEB), and inhibit the expression of autophagy-related genes, thereby suppressing the normal level of autophagy and promoting apoptosis, which in turn leads to cellular aging (Roczniak-Ferguson et al., 2012; Wang et al., 2024; Yadav et al., 2018). In addition, AhR has been found to activate the expression of downstream target gene CYP1A1. Polychlorinated biphenyls (PCBs) have been observed to increase the expression level of AhR in mouse testis, and AhR has been found to target CYP1A1 to promote the synthesis of cytochrome oxidase P450, generating a large amount of ROS to increase the level of oxidative stress and inducing apoptosis (Wang et al., 2023). Consequently, the effective blockade of the Kyn-AhR pathway emerges as a promising approach for addressing age-related diseases and reproductive system damage. Blocking Kyn-AhR-induced anabolic disorders in the body is also an important key to preventing and treating reproductive diseases. It has been found that after treatment of granulosa cells with kynurenine, kynurenine can inhibit progesterone biosynthesis in granulosa cells through the Kyn-AhR pathway by down-regulating the expression of GATA4, GATA6, and CEBPB, and thus lowering the estrogen level of the organism (Liao et al., 2025). The Kyn-AhR axis also acts on the PCSK9 pathway, thereby mediating an increase in LDL-C production and other lipid metabolism abnormalities, leading to pathological changes in female PCOS (see in ‘Kyn and the Reproductive System’). Research on AhR antagonists is currently in its nascent stages. However, CB7993113 has been reported to effectively block AhR in vitro, inhibit TNBC cell migration and invasion, and reduce the metastatic potential of breast cancer cells (Parks et al., 2014). The substantial accumulation of Kyn in reproductive-related diseases such as polycystic ovary syndrome and ovarian failure (Shen et al., 2023; Wang et al., 2022b) suggests that the combined application of IDO or TDO inhibitors with AhR antagonists could be a potential mechanism for combating reproductive disorders. However, further investigation is necessary to fully elucidate the efficacy and potential applications of this approach.

Prevention of KP metabolite-mediated oxidative stress

Metabolites of the kynurenine pathway have been demonstrated to be associated with oxidative stress. 3-hydroxykynurenine (3-HK) and its cleavage product, 3-hydroxy-o-cyclo-aminobenzoic acid (3-HAA), which is produced by Kyn catalyzed by KMO, have been shown to mediate oxidative stress that significantly damages neuronal tissues (Bonda et al., 2010). Furthermore, Kyn has been shown to promote the synthesis of cytochrome oxidase P450 through the activation of the expression of the downstream target gene, CYP1A1, via AhR. synthesis, which generates large amounts of superoxide anion to increase oxidative stress (OS) levels (Wang et al., 2023). Increased OS levels adversely affect the reproductive system in both sexes. In males, elevated OS levels can result in the onset of inflammatory conditions within the reproductive system, such as orchitis, which can lead to damage to reproductive cells, including spermatozoa, and is associated with male infertility (Dutta et al., 2021). The underlying mechanism may involve the damage to the DNA structure of spermatozoa by supraphysiologic ROS levels, thereby affecting their normal development and survival (Lopes et al., 2021). Furthermore, the impact of ROS on the hypothalamic-pituitary-gonadal axis, a critical regulatory system in the body, can lead to a reduction in testosterone production and the exacerbation of male infertility (Rotimi et al., 2024). In female subjects, the deleterious effects of OS on the reproductive system primarily manifest as damage to oocytes, a process that is associated with the aging of the ovaries and the development of polycystic ovary syndrome (Gongadashetti et al., 2021; Smits et al., 2023). Therefore, regulating the production of Kyn and 3-HK in the KP pathway to reduce OS levels is crucial for the prevention and treatment of reproductive disorders. Currently, resveratrol is the most likely potential mechanism with targets of action that could link Kyn metabolism to OS. Resveratrol is a polyphenol with anti-aging and ovarian senescence prevention properties that is widely used. In a study of the chicken gut microbiota, resveratrol was found to be effective in ameliorating the decline in ovarian function and improving fertility by modulating the KP pathway and thereby reducing OS levels (Wang et al., 2022a). Melatonin, another end product of tryptophan metabolism, has antioxidant functions in addition to the KP pathway (i.e., metabolized via the 5-HT pathway). It has been shown that melatonin can exert antioxidant effects through the MT-1 receptor pathway, thereby preventing ovarian damage (Barberino et al., 2017). Similarly, melatonin can mitigate OS damage to DNA in a SIRT1-dependent manner and play a testicular protective role (Xu et al., 2020). Consequently, increasing melatonin production by inhibiting the KP emerges as a promising mechanism for the prevention and treatment of reproductive system diseases.

Supplementation of NAD precursors

The interconnected nature of metabolic pathways and processes renders them susceptible to disruption, which underscores the necessity for a comprehensive understanding of the KP pathway. In addition to the potentially deleterious byproducts of metabolic processes, the beneficial end-product of the KP pathway, NAD, and the potential inhibition of the KP pathway for the prevention and treatment of diseases warrant consideration. The inhibition of the de novo NAD synthesis pathway (see at section ‘Introduction’) has been implicated in the onset of reproductive disorders mentioned in sections ‘Regulation of key enzyme activities of KP’, ‘Inhibition of the Kyn-AhR pathway’, and ‘Prevention of KP metabolite-mediated oxidative stress’, underscoring the necessity for vigilant monitoring. The deleterious effects of diminished NAD production on the reproductive system and the body as a whole can be mitigated through prompt supplementation with NAD or its precursors. The maintenance of optimal NAD levels is imperative for the proper functioning of the reproductive system. It has been observed that NAD levels are diminished in senescent oocytes, leading to impediments in normal maternal reproductive function (Di Emidio et al., 2024). Conversely, augmenting ovarian NAD levels has been shown to enhance mitochondrial function and even reverse the onset of ovarian senescence (Yang et al., 2020), thereby promoting reproduction. Furthermore, studies on male mice have revealed that low NAD levels can trigger a decline in testicular function and disruption in sperm development, which is a significant mechanism contributing to reduced male fertility (Meyer-Ficca et al., 2022). Consequently, the supplementation of NAD or its precursors in conjunction with the inhibition of the KP pathway has emerged as a crucial approach to address these issues. Nicotinamide mononucleotide (NMN), a precursor of NAD, has been shown to enhance the quality of senescent oocytes, promote ovulation, and augment their meiotic ability and fertilization potential, thereby contributing to improved fertility (Miao et al., 2020). In addition, supplementation with NAD precursors such as NMN and nicotinamide riboside (NR) can improve polycystic ovary syndrome by effectively alleviating insulin resistance and reducing inflammatory stress to restore mitochondrial function (Aflatounian et al., 2022; Wang et al., 2021). In men, supplementation with NAD precursors can inhibit the Sirt2 pathway, thereby protecting sperm from iron death attack and ameliorating male reproductive disorders (Feng et al., 2024). However, it is worth noting that increased levels of NAD synthesis are considered to be a marker of tumor cell proliferation (Ghanem, Monacelli & Nencioni, 2021), so whether NAD precursors can be applied in reproductive system tumors remains to be further explored.

KP modulation in combination with immunotherapy

IDO can promote apoptosis of T cells and DC cells to exert immunosuppressive effects and promote tumor growth (see ‘IDO and the Reproductive System’). Therefore, the combination of modulation of KP metabolism and immunotherapy has become a new idea to fight against reproductive system tumors. It has been demonstrated that under the combined treatment using IDO1 inhibitor and immune checkpoint blockade (ICB), the activity of highly expressed IDO1 was blocked, thereby effectively reducing the immune escape of cervical cancer cells and decreasing the tumor size, and the effect of the combined treatment was significantly more effective than the application of ICB alone (Qu et al., 2023). In addition to IDO, TDO also plays an important role in tumor progression. The dual IDO/TDO inhibitor AT-0174 disrupted the immunosuppressive tumor microenvironment of ovarian cancer patients, reduced tumor-associated macrophage infiltration, and inhibited ovarian cancer progression. In addition, when combined with chemotherapy, patient survival was prolonged (Crump et al., 2024). Notably, the antitumor effect of IDO inhibitors alone was not significant, with a disease stabilization rate of only 50%, whereas 80% of patients were stabilized after combination with the PD-L1 inhibitor atezolizumab (Ebata et al., 2020). However, KP modulation in combination with immunotherapy does not seem to be suitable for all tumor types. In a phase III clinical trial for the treatment of melanoma, the combination of the IDO inhibitor epacadostat did not provide greater clinical efficacy compared to pembrolizumab alone (Long et al., 2018). The reason for this may be due to differences in the expression levels of IDO or TDO in different tumors or the low selectivity of inhibitors for tumor IDO. For example, depletion of Kyn with polyethylene glycolated kynureninase in 4TA breast cancer cells with high expression of IDO1 could reverse the upregulation of IDO1 in the tumor microenvironment and thus inhibit tumor growth (Triplett et al., 2018). Therefore, the combination of KP modulation and immunotherapy is suitable for IDO/TDO high-expressing reproductive system tumors and the selection of appropriate inhibitors. However, to date, no KP markers other than IDO/TDO have been included in the study of immunotherapy combination, and there are few studies on male reproductive system tumors, plus the research on the combination of IDO inhibitors and immunotherapy is not yet mature, so the research on the application of this method is still full of challenges. However, in conclusion, tumor suppression by KP modulation and immune checkpoint inhibition is a promising potential approach in the treatment of reproductive system tumors.

KP regulation of gut flora

A numerous of studies in recent years have demonstrated that the prevention and treatment of diseases of the reproductive system should not only focus on the reproductive system itself, but also on other systems of the body. For instance, the impact of intestinal flora on reproductive health is an aspect that is often overlooked. The gut flora has the capacity to modify the intestinal environment through metabolic processes and other life activities, which in turn affects organs or systems outside the gut. This suggests that the gut flora can be considered an endocrine organ (Qi et al., 2021), and therefore dysregulation of gut microbial homeostasis can adversely affect the male and female reproductive systems. Imbalanced population ratios of gut flora have been found to cause hormonal metabolic imbalances, including decreased oestrogen, increased testosterone, and insulin resistance. These imbalances have been associated with the development of endometriosis and polycystic ovary syndrome (Lüll et al., 2021; Plottel & Blaser, 2011). Furthermore, alterations in gut flora have been associated with the risk of male infertility (Fu, Wang & Yan, 2023). Abnormalities in tryptophan-kynurenine metabolism in gut microbes play an important pathological role in extraintestinal organs, and KP becomes a pathogenic crossroads linking gut and extraintestinal organ disorders (Wiedlocha et al., 2021). Similarly, gut flora’s KP imbalances have detrimental effects on the reproductive system. Metabolic activation of the gut flora-associated KP can lead to elevated levels of ovarian oxidative stress, resulting in decreased ovarian function and worsened fertility levels. However, oral administration of resveratrol has been shown to reverse these pathological effects (Wang et al., 2022a). Similarly, KP regulation of intestinal flora plays an active role in the prevention of HPV-associated cervical cancer. A clinical trial found that oral administration of probiotics significantly down-regulated IDO and TDO expression to attenuate kynurenine pathway activity and reduce stress compared to the control group. This ultimately resulted in increased immunity and decreased vaginal HPV abundance in HPV-positive women, thereby reducing the risk of cervical cancer (Xu et al., 2025). It is evident that regulating KP metabolism in the gut flora can positively affect the reproductive system. However, at present, there are few studies on the crosstalk between abnormal tryptophan metabolism in the gut flora and the reproductive system, and further research is required to explore the direct link between reproductive health and KP in the gut flora. In summary, maintaining good gut ecology and regulating gut flora KP are highly promising mechanisms for maintaining reproductive health as well as preventing and controlling reproductive system diseases. A healthy lifestyle, including regular running exercise in moderation, has been shown to modulate the ability of gut microbes to metabolise tryptophan and to increase the transport of tryptophan to the hippocampus and brainstem (Vazquez-Medina et al., 2024). This may have potential benefits for neurological and psychological health. Conversely, the long-term consumption of a high-fat diet has been demonstrated to induce an imbalance in the composition of gut microbiota, leading to an alteration in the metabolism of tryptophan within colonic cells and an upregulation of IDO expression. This, in turn, results in a depletion of Kyn within the intestine and an accumulation of Kyn in the peripheral bloodstream (Sun et al., 2023), which may potentially impact reproductive health. In conclusion, there is an urgent need for in-depth research on the mechanism of reproductive system disease prevention and control, with a focus on the maintenance of healthy lifestyle habits to regulate intestinal flora’s KP.

Controversies and Knowledge Gaps

As mentioned earlier, the role of the tryptophan-kynurenine pathway in disease is complex, and the exact pathological mechanisms are still not well understood. For example, two diametrically opposed effects of KYNU on breast cancer promotion and inhibition have been reported (see ‘KYNU and the Reproductive System’). The mechanism of KYNU promotion of breast cancer may be related to the adenosine deaminase acting on RNA 1 (ADAR1)-KYNU interaction. The expression levels of both KYNU and ADAR1 were found to be upregulated in breast cancer cells, and an interaction between the two was detected by immunoprecipitation, while subcellular localization showed that KYNU was predominantly enriched in the nuclear periphery, whereas ADAR1 was located predominantly in the nucleus and cytoplasm of the cells (Binothman et al., 2023). It is hypothesized that KYNU-ADAR1 interaction may affect the expression of proto-oncogenes or oncogenes, thereby promoting breast cancer progression. Interestingly, the inhibitory effect of KYNU on breast cancer is importantly related to the expression of IDO1/TDO2. In breast cancer patients with high expression of IDO1/TDO2, low expression of KYNU was associated with a higher degree of tumor infiltration and lymphatic infiltration, whereas high expression of KYNU was significantly associated with a better level of prognosis in the patients. Co-expression of KYNU seems to play a role in the elimination of immunosuppressive tumor-promoting effects of IDO1/TDO2 (Giatromanolaki et al., 2025). Therefore, other key enzymes of the tryptophan-kynurenine pathway should be considered together when discussing the pathogenic role of KYNU in reproductive diseases as a way to identify potential therapeutic targets.

Summary and Prospects

The Trp-Kyn metabolic pathway is an important part of tryptophan metabolism and has significance in numerous reproductive disorders. This review reviews and summarizes the mechanism and relevance of various key enzymes and downstream products of the Trp-Kyn metabolic pathway in reproductive diseases, and explores the potential mechanisms of targeting the kyn pathway for the treatment of reproductive diseases. However, the metabolic pathway of tryptophan is complex and delicate, and the intermediate products or enzymes have different or even diametrically opposite effects on the human body, so we still need to travel a long way from the mechanism study to clinical application. For example, IDO mediates relevant cancers such as breast cancer and ovarian plasma adenocarcinoma through the Trp-AhR pathway, while in the testis, it mediates testicular immune privilege, preventing the body’s immune cells from attacking their own antigens. A wide range of intermediates of KP, as well as its key enzymes, play different roles in different reproductive system diseases. In conclusion, KP causes reproductive system-related diseases that can be categorized into five main groups: (1) Inhibition of immune cell activity and promotion of the formation of a tumor microenvironment. (2) Induction of inflammation. (3) Raising the level of oxidative stress to promote reproductive damage and senescence. (4) Inhibition of the level of normal autophagy and mediation of the onset of reproductive senescence. (5) Impairment of reproductive health through metabolic abnormalities of the gut flora. Therefore, KP may constitute a new mechanism for the prevention and treatment of reproductive system-related diseases. Currently popular studies of potential mechanisms include inhibitors of various key enzymes of KP, regulation of KP metabolites, and regulation of KP in the gut flora. The molecular mechanisms of KP intermediates in the development of reproductive system-related diseases are still not fully defined and need to be explored further. The interaction of the kynurenine pathway with cell signaling pathways such as autophagy or apoptosis can be explored in depth. It is also possible to look at the reproductive “secondary pathologies” that result from the “primary pathology” of abnormal kynurenine metabolism in various systems throughout the body. (e.g., crosstalk between abnormal tryptophan metabolism in the intestinal flora and the reproductive system). For example, whether the aberrant expression of KP intermediates alters the function of related organs or organelles, thus leading to reproductive system diseases and their underlying molecular mechanisms, should be further investigated. Currently, cellular senescence is a hot topic, and investigations of the regulatory role of KP in the senescence process are promising and are conducive to obtaining a better understanding of the mechanism of KP in related diseases. However, the relationship between KP and organelle dysfunction remains to be explored. Currently, our group is investigating the mechanism related to Kyn and mitochondrial autophagy leading to aging, which may provide new insights for the prevention and treatment of reproductive system-related diseases. (A brief summary of this review is listed at Fig. 4).

Figure 4 Summary of KP in reproductive system-related diseases and possible treatment mechanisms.

(Brincks et al., 2020; Sieviläinen et al., 2022; Gibney et al., 2019; Chuang et al., 2024; Oweira et al., 2022; Rogers et al., 2019; Xue et al., 2023; Hughes et al., 2022; Bonda et al., 2010; Wang et al., 2023; Di Emidio et al., 2024; Qu et al., 2023; Wang et al., 2022a).

Supplemental Information

Supplemental Information 1 PRISMA flowchart

Abbreviations

Trp Tryptophan

KP Kynurenine pathway

TDO Tryptophan 2,3-dioxygenase

IDO Indoleamine 2,3-dioxygenase

Kyn Kynurenine

KMO Kynurenine-3-monooxygenase

3-HK 3-Hydroxykynurenine

KYNU Kynureninase

3-HAA 3-Hydroxyanthranilic acid

QUIN Quinolinic acid

QPRT Quinolinic acid phosphoribosyltransferase

KAT Kynurenine aminotransferase

KYNA Kynurenic acid

NAD Nicotinamide adenine dinucleotide

TNBC Triple-negative breast cancer

AhR Aryl hydrocarbon receptor

ET Endometriosis

BC Breast cancer

ADC Cervical adenocarcinoma

CCBL2 Cysteine-conjugated β-cleavage enzyme

PCOS Polycystic ovary syndrome

OS Oxidative stress

ROS Reactive oxygen species

Additional Information and Declarations

Competing Interests

Author Contributions

Data Availability

The authors declare there are no competing interests.

Zhongkai Ou conceived and designed the experiments, performed the experiments, analyzed the data, prepared figures and/or tables, authored or reviewed drafts of the article, and approved the final draft.

Aixia Xu performed the experiments, authored or reviewed drafts of the article, and approved the final draft.

Hua Su analyzed the data, authored or reviewed drafts of the article, and approved the final draft.

Yiting Liu performed the experiments, prepared figures and/or tables, and approved the final draft.

Jia Li conceived and designed the experiments, authored or reviewed drafts of the article, resources, Supervision, Funding acquisition, Project administration, and approved the final draft.

The following information was supplied regarding data availability:

This article is a literature review.

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
