# Peer review of "Tryptophan-kynurenine pathway: a possible new mechanism for the prevention and treatment of reproductive system-related diseases"

_PeerJ, doi:10.7717/peerj.20342_

## Round 0.1 · original submission · Major Revisions

· Academic Editor

Major Revisions

Dear authors,

Please refer to the reviewers' comments for further details.

Reviewer 1 ·

Basic reporting

Replace vague phrases such as “and so on” with the exact search terms used to enhance transparency and reproducibility.
Line 160: Revise “poor prognostic marker” to “good prognostic marker for poor survival.”
Include a discussion on the relevance of the tryptophan depletion theory in cancer progression, as this is a critical and widely recognised mechanism.
Figure 2: Correct the spelling of "consumption."
Line 213: Correct “TO2” to “TDO2.”
Lines 243 and 450: Use consistent terminology. “Tryptophan” should be abbreviated consistently (e.g., Trp), and abbreviations should be defined at first mention.
Line 244: The phrase “in conclusion” appears in the middle of the manuscript, which is confusing. Please rephrase or relocate this to the conclusion section.
Line 247: Consider rewording “promote tumour environment” to “supportive” or “enabling tumour environment” for greater clarity and precision.
Aromatase expression is briefly mentioned but not discussed in depth. Expanding on its relevance in the context of the TRP/KYN pathway or reproductive diseases would enhance the manuscript.
Line 294: Remove the phrase “by investigators” as it is unnecessary.
Line 306: Provide context for what the T stage entails to aid reader comprehension, particularly for a multidisciplinary audience.
The manuscript inconsistently refers to NAD and NAD+. Please standardize this throughout.
Figure 3: Consider the key message of this figure. Currently, it duplicates the same pathway in two cells, which may not be necessary. You might remove the redundant cell and instead use a note to indicate potential autocrine or paracrine signaling, if that is the intended message.
Line 256: Drug names should not be capitalised unless they begin a sentence.
Line 461: Define CBP prior to its first use.
Lines 735–747 (Conclusion): This section is overly general and does not clearly address the manuscript's focus on reproductive diseases. Rework the conclusion to tie back more directly to this topic.
Consider including a summary table outlining key metabolites or enzymes and their effects across various diseases. This would greatly enhance readability and provide a quick reference for the audience.

Experimental design

The manuscript is thought-provoking and well-written.
However, in the first half, the content appears to be predominantly focused on cancer. It is recommended that the title be revised to reflect this emphasis, or alternatively, that additional information on other diseases be included to create a more balanced overview. Given the abundance of existing reviews on cancer, readers would likely benefit more from expanded discussion on reproductive and other diseases. In particular, incorporating more detail on current treatments for reproductive conditions, and how these treatments interact with the TRP/KYN pathway, would be valuable. The discussion of resveratrol serves as a strong example in this regard.

Validity of the findings

It would be valuable to elaborate on future directions based on the findings.

·

Basic reporting

Dear my honored Editor in Chief,
It is my great honor to review the paper form your Journal Peer J.
General Comments:
The submitted manuscript (Peer J-117671) is about the “Tryptophan-kynurenine pathway: a possible new mechanism for the prevention and treatment of reproductive system-related diseases" has been reviewed. This comprehensive review synthesizes current knowledge on the role of the tryptophan-kynurenine pathway (KP) in reproductive system diseases and its therapeutic potential. The topic is timely, clinically relevant, and aligns with PeerJ's scope. The authors effectively highlight the dual roles of KP enzymes/metabolites in pathologies (e.g., cancer, endometriosis) and physiological functions (e.g., testicular immune privilege). However, the manuscript requires significant revisions to enhance clarity, address structural redundancies, update literature, and strengthen critical analysis. Major Concerns
1.The Abstract and Introduction (pages 4–6) are nearly identical. This redundancy dilutes impact.Condense the Introduction to focus on knowledge gaps and the review’s objectives. The Abstract should succinctly summarize key findings/therapeutic implications.

2. Contradictory findings (e.g., KYNU’s tumor-suppressive vs. promotive roles in breast cancer) are noted but not critically evaluated. The underlying mechanisms (e.g., tissue-specific expression, signaling crosstalk) need deeper discussion. Add a subsection (e.g., "Controversies and Knowledge Gaps") to explore these inconsistencies and propose research directions.
3. Key citations predate 2020 (e.g., Okamoto et al. 2005, Wang et al. 2012). Recent breakthroughs (e.g., KP modulation in immunotherapy trials, NAD+precursors in clinical studies) are underrepresented. Include the latest literature about ovarian aging and KP gut-microbiome crosstalk during 2023-2025.
4. Section 3 ("Possible Treatment Mechanisms") overemphasizes preclinical data without addressing clinical challenges (e.g., inhibitor toxicity, biomarker validation).
Discuss ongoing clinical trials (e.g., NCT03343613 for IDO inhibitors in ovarian cancer). Address safety concerns (e.g., risks of systemic KP inhibition in non-target tissues).

5. Figure 2: Fails to distinguish IDO’s pathological vs. protective roles. Use color coding or split panels for clarity.

6.Figure 4: "KP regulation of gut flora" lacks mechanistic detail. Add arrows linking gut metabolites (e.g., Kyn) to reproductive organs.

7.Tables: Include a summary table of KP enzymes/metabolites, their roles in specific diseases, and associated therapeutics.

Minor Concerns
Page 6, Line 75: "Quinolinic acid is a neurotoxic metabolite in the KP(12)" change to "Quinolinic acid (QUIN), a neurotoxic KP metabolite (12), [...]".
Page 10, Line 241: "Kyn is a specific player" change to "Kynurenine (Kyn) is a key mediator".
Inconsistent abbreviation use (e.g., "Trp" vs. "Try"; "AhR" vs. "AHR"). Define all abbreviations at first use (e.g., Page 5: "tryptophan (Trp)") and maintain consistency.
The search methodology (Page 7) lacks critical details (e.g., search dates, PRISMA flow).

Experimental design

This is a review.

Validity of the findings

This comprehensive review synthesizes current knowledge on the role of the tryptophan-kynurenine pathway (KP) in reproductive system diseases and its therapeutic potential. The topic is timely, clinically relevant, and aligns with PeerJ's scope. The authors effectively highlight the dual roles of KP enzymes/metabolites in pathologies (e.g., cancer, endometriosis) and physiological functions (e.g., testicular immune privilege). However, the manuscript requires significant revisions to enhance clarity, address structural redundancies, update literature, and strengthen critical analysis.

Additional comments

Dear my honored Editor in Chief,
It is my great honor to review the paper form your Journal Peer J.
General Comments:
The submitted manuscript (Peer J-117671) is about the “Tryptophan-kynurenine pathway: a possible new mechanism for the prevention and treatment of reproductive system-related diseases" has been reviewed. This comprehensive review synthesizes current knowledge on the role of the tryptophan-kynurenine pathway (KP) in reproductive system diseases and its therapeutic potential. The topic is timely, clinically relevant, and aligns with PeerJ's scope. The authors effectively highlight the dual roles of KP enzymes/metabolites in pathologies (e.g., cancer, endometriosis) and physiological functions (e.g., testicular immune privilege). However, the manuscript requires significant revisions to enhance clarity, address structural redundancies, update literature, and strengthen critical analysis. Major Concerns
1.The Abstract and Introduction (pages 4–6) are nearly identical. This redundancy dilutes impact.Condense the Introduction to focus on knowledge gaps and the review’s objectives. The Abstract should succinctly summarize key findings/therapeutic implications.

2. Contradictory findings (e.g., KYNU’s tumor-suppressive vs. promotive roles in breast cancer) are noted but not critically evaluated. The underlying mechanisms (e.g., tissue-specific expression, signaling crosstalk) need deeper discussion. Add a subsection (e.g., "Controversies and Knowledge Gaps") to explore these inconsistencies and propose research directions.
3. Key citations predate 2020 (e.g., Okamoto et al. 2005, Wang et al. 2012). Recent breakthroughs (e.g., KP modulation in immunotherapy trials, NAD+precursors in clinical studies) are underrepresented. Include the latest literature about ovarian aging and KP gut-microbiome crosstalk during 2023-2025.
4. Section 3 ("Possible Treatment Mechanisms") overemphasizes preclinical data without addressing clinical challenges (e.g., inhibitor toxicity, biomarker validation).
Discuss ongoing clinical trials (e.g., NCT03343613 for IDO inhibitors in ovarian cancer). Address safety concerns (e.g., risks of systemic KP inhibition in non-target tissues).

5. Figure 2: Fails to distinguish IDO’s pathological vs. protective roles. Use color coding or split panels for clarity.

6.Figure 4: "KP regulation of gut flora" lacks mechanistic detail. Add arrows linking gut metabolites (e.g., Kyn) to reproductive organs.

7.Tables: Include a summary table of KP enzymes/metabolites, their roles in specific diseases, and associated therapeutics.

Minor Concerns
Page 6, Line 75: "Quinolinic acid is a neurotoxic metabolite in the KP(12)" change to "Quinolinic acid (QUIN), a neurotoxic KP metabolite (12), [...]".
Page 10, Line 241: "Kyn is a specific player" change to "Kynurenine (Kyn) is a key mediator".
Inconsistent abbreviation use (e.g., "Trp" vs. "Try"; "AhR" vs. "AHR"). Define all abbreviations at first use (e.g., Page 5: "tryptophan (Trp)") and maintain consistency.
The search methodology (Page 7) lacks critical details (e.g., search dates, PRISMA flow).

---

## Round 0.2 · accepted · Accept

· Academic Editor

Accept

Dear authors,

All issues have now been resolved. Thank you for your submission and hard work. I am now accepting your work for publication in PeerJ. Congratulations.

Reviewer 1 ·

Basic reporting

no comment

Experimental design

no comment

Validity of the findings

no comment

Additional comments

The authors have adequately addressed the comments, and I recommend this manuscript for publication.